# Faba Bean Flavor Effects from Processing to Consumer Acceptability

**DOI:** 10.3390/foods12112237

**Published:** 2023-06-01

**Authors:** Abraham Badjona, Robert Bradshaw, Caroline Millman, Martin Howarth, Bipro Dubey

**Affiliations:** 1National Centre of Excellence for Food Engineering, Sheffield Hallam University, Sheffield S1 1WB, UK; a.badjona@shu.ac.uk (A.B.);; 2Bimolecular Research Centre, Sheffield Hallam University, Sheffield S1 1WB, UK

**Keywords:** faba bean, ‘beany’ off-flavor, processing effects, consumer acceptance, aroma, product preparation

## Abstract

Faba beans as an alternative source of protein have received significant attention from consumers and the food industry. Flavor represents a major driving force that hinders the utilization faba beans in various products due to off-flavor. Off-flavors are produced from degradation of amino acids and unsaturated fatty acids during seed development and post-harvest processing stages (storage, dehulling, thermal treatment, and protein extraction). In this review, we discuss the current state of knowledge on the aroma of faba bean ingredients and various aspects, such as cultivar, processing, and product formulation that influence flavour. Germination, fermentation, and pH modulation were identified as promising methods to improve overall flavor and bitter compounds. The probable pathway in controlling off-flavor evolution during processing has also been discussed to provide efficient strategies to limit their impact and to encourage the use of faba bean ingredients in healthy food design.

## 1. Introduction

Increasing market demand for plant-based ingredients as alternatives to animal-based materials has been receiving growing attention over the past decades. Hence, several investigations are being conducted on the sensory properties of plant-based ingredients, such as peas and faba beans [1,2]. In addition, the current disparity between global food supply and demand for meat-based products is becoming a rising concern due to the recent coronavirus disease outbreak (COVID-19) that affected the food supply chain [3]. Concerns regarding environmental sustainability and protection of animals, as well as health-related benefits, have also increased the recent interest in plant-based ingredients [4,5]. Faba bean is a legume-based protein that contains approximately 250 g protein/kg seed. Approximately 43,000 faba bean accessions are held in the GenBank database globally [6]. Storage proteins in faba beans are made up of the 11S (legumin) and 7S (vicilin) proteins, each of which has unique technological, functional, and physicochemical characteristics [7,8]. According to studies, faba bean proteins are equivalent to most high-quality plant-based proteins, such as peas and soy, and, therefore, represent alternate protein sources [9]. 

Despite the nutritional, health, and agronomical benefits associated with faba beans, utilization of their ingredients has been hindered by the presence of antinutritional factors and undesirable sensory attributes, such as colour and off-flavor. As a multi-component system, faba beans include a number of flavor-formation precursors, making it difficult to generate a consensus on the compounds responsible for beany flavor in faba bean ingredients. Characterization and identification of volatile aroma compounds is thus useful for selecting and commercialising faba bean cultivars for specific food applications. Flavor is an important sensory characteristic that influences consumer acceptance of foods. Flavor perception is mostly modulated by organoleptic properties, such as aroma, texture, taste, and mouthfeel [10]. A major drawback in the development of novel plant-based foods is the off-flavor associated with them during incorporation into traditional foods. Consumer acceptance of alternative meat products is highly dependent on their healthiness and sensory properties, as observed by [11]. According to research of customers from Germany, France, and the UK, those who eat meat have low expectations for the flavor of meat substitutes. They anticipated that a plant-based burger would be less preferable than a beef burger [12]. To limit these drawbacks, flavor, as well as functionality improvement, will be useful means of increasing direct use in human foods. In this regard, increasing the consumption of faba bean ingredients in different products will require careful consideration of organoleptic properties, such as flavor and taste, to meet consumer expectations during processing and product design. The potential application of faba beans in food is more complex than what first appears. Faba beans are frequently transformed into flour, concentrate, and isolates and can undergo modification during processing to improve their application during food systems [13,14]. It is possible to generate pasta, crackers, flakes, mayonnaise, and dairy or meat analogues using the functional ingredients derived from the bean itself, such as flour or protein isolate and concentrate [15,16,17]. During processing along the food value chain, off-flavor compounds may be reduced or conversely limit the presence of undesirable flavor compounds [18]. 

This review provides a general overview of faba bean flavor, and key beany flavor is discussed, as well as its origin, is also investigated. Effects of processing conditions on faba bean derived products volatiles are also reviewed. In addition, various recommendations and future research areas are also identified and discussed. 

## 2. Volatile Odorant Compound

Volatile compounds are low molecular weight organic compounds (<300 Daltons) with strong hydrophobicity that can reach the olfactory receptors due to their high volatility [19,20]. They comprise ketones, aldehydes, alcohols, carboxylic acid, terpenes, sulphur-containing compounds, aromatic hydrocarbons, and methoxypyrazines [19,21]. The onset of volatile generation begins during seed development and evolves during harvesting, processing, and storage. Production of volatiles is usually intensified when pulses are subjected to intense stressful conditions, such as temperature, mechanical damage, as well as insect infestation and water stress [22]. Hence, their production is often a defense response against adverse conditions [23].

The flavor characteristics of faba beans are generally affected by type of cultivar, storage conditions, processing, and geographical location [24]. One major analytical technique that has been useful for the investigation of volatile odorant compounds is the use of headspace solid-phase micro-extraction (HS-SPME) with gas chromatography coupled to mass spectrometry (GC–MS) [25,26]. For routine investigation of the volatile profile of faba beans, solid-phase micro-extraction technology has shown to be a quick and efficient approach that may be used in research [27]. Faba bean contains a total of 36 volatile compounds, including monoterpene hydrocarbons, oxygenated monoterpenes, sesquiterpene hydrocarbons, phenylpropanoids, apocarotenes, and non-terpene derivatives, according to research by [28].

Depending on the analytical procedure employed, as well as the SPME fibre used, the aroma profile may vary considerably. However, ref. [27] revealed the presence of 45 volatile compounds in different faba beans cultivars consisting of aldehydes, aromatic hydrocarbons, alcohols, ketones, and alkanes. Aldehydes made up on average 57%, with aromatic hydrocarbons, alcohols, ketones, alkanes, furans, and alkenes following in decreasing percentages. Aldehydes were rather abundant and may have a substantial impact on the aroma of faba beans. The volatiles revealed by [27,28] showed fewer volatile compounds compared to Canadian dry faba beans, which contained 63 volatiles [26]. However, it was observed that 29 of these compounds were present in both beans. In the same instance, aldehydes were the most dominant group present in faba beans, while, in dry beans, the main class was aromatic hydrocarbons. Volatile compounds, such as 1-hexanol, have been demonstrated to repel insects, indicating their abundance and role in preventing insect and/or pathogen infestation acting as a defence mechanism, as stated earlier [29,30].

Pentanal, hexanal, heptanal, octanal, nonanal, decanal, 1-hexanol, and 1-octen-3-ol are among the 33 key volatile compounds (ROAV ≥ 0.1) that contribute to the flavor of raw faba bean flour as shown in Table 1. These compounds likely result from the oxidation of unsaturated fatty acids, while 3-methylbutanal, phenylacetaldehyde, and 3-methylbutyric acid are possibly by-products of amino acid degradation. The remaining four volatiles, which include D-limonene, -linalool, menthol, and estragole, are often found volatiles and may originate from many sources [31]. Any volatile compound that has a low threshold of detection expressed in parts per billion (ppb) is often sufficient to evoke a sensorial flavor response [32].

Precursors of volatile compounds are usually present in different forms in faba beans and are carried over into concentrate and isolate following processing and storage. Oxidative degradation of unsaturated fatty acid has been attributed to be the major contributor to volatile compound generation, especially off-flavors. Hulled and whole faba bean seeds contain approximately 2–3% of lipids [33,34]. Fatty acids, such as linoleic, linolenic, and other fatty acids, make up 56% of the total fatty acids present in faba bean. Unsaturated fatty acid accounts for 80.80% of the total fatty acid present in faba beans [34]. Fatty acids are usually present as phospholipids or in esterified form as triglycerides; however, few amounts may be present in the free form. Pre-treatments, processing, and storage conditions make polyunsaturated fatty acids susceptible to oxidative degradation. Fatty acid degradation is initiated and promoted through lipid modifying enzymes, such as lipoxygenase, oxidation (atmospheric oxygen), photo-oxidation (singlet oxidation), and autoxidation in the presence of suitable catalysts, all of which generate hydroperoxides as the main products and subsequently degrade to produce volatile compounds [35,36]. The main volatile compounds resulting from the degradation of lipids include aldehydes, alcohols, esters, and ketones. These volatiles tend to impact aroma at even low concentrations, and, for instance, the flavors of aldehydes (hexanal and octanal) are in the range of 1–5 ppb [32].

Despite the single role each volatile compound elicits as a specific odour, it is imperative to note that, due to the complexity of different chemical groups, their concentrations, as well as the influence of human perception on flavor, it is more difficult to solve flavor issues in complex systems, since individual volatile compound alone does not explain its impact on flavor characteristics of a product. Hence, to explore flavor complexity, including specific flavor compounds, a novel approach, such as untargeted metabolomic, can be employed using GC–MS and GC–Olfactory measurements [37].

**Table 1 foods-12-02237-t001:** Major off-flavor compounds associated with faba bean ingredients, their chemical structure and odour threshold (OT/ppb) values. Note: Odor thresholds in water from ^a^ [38]; ^b^ [39]; ^c^ [40]; ^d^ [41]; ^e^ [42]; ^f^ [43].

Odorant	CAS Registry Number	Chemical Structure	OT/ppb	Aroma Attributes	Flour [44]	Concentrate [45]
**Aldehyde**						
(E)-2-hexenal	6728-26-3	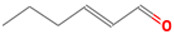	0.082 ^ab^	leafy	−	**+**
(E)-2-heptenal	18829-55-5	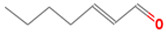	0.013 ^ab^	soap, fat	−	**+**
(E)-2-octenal	2548-87-0	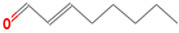	0.003 ^ab^	fatty, green, cucumber	**+**	**+**
(Z)-3-hexenal	6789-80-6	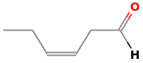	0.00012 ^a^	green	−	−
pentanal	110-62-3	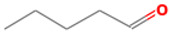	0.008 ^ac^	irritant	−	**+**
hexanal	66-25-1	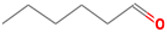	0.0045 ^ac^	cut-grass, green	**+**	**+**
heptanal	111-71-7	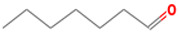	0.003 ^ac^	dry fish	**+**	**+**
nonanal	124-19-6	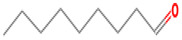	0.001 ^ac^	green, fatty	**+**	**+**
octanal	124-13-0	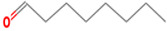	0.0007 ^ac^	fatty, pungent	**+**	**+**
(E)-2-nonenal	18829-56-6	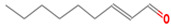	0.0004 ^ad^	beany, green	**+**	**+**
benzaldehyde	100-52-7	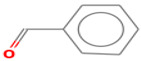	0.003 ^ac^	almond	**+**	−
(E, E)-2,4- nonadienal	5910-87-2	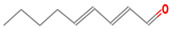	0.0001 ^a^	fatty	−	**+**
(E, E)-2, 4-heptadienal	4313-03-5	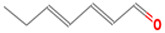	0.00256 ^a^	fatty, fishy	−	**+**
(E,E)-2,4 decadienal	25152-84-5	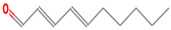	0.000027 ^a^	spices	−	**+**
**Pyrazine**						
2-isopropyl-3- methoxypyrazine	25773-40-4	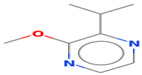	0.0008 ^ef^	pea-like, earthy	−	−
**Acid**						
acetic acid	64-19-7	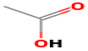	180 ^ad^	sour	**+**	**+**
**Ketone**						
2-heptanone	110-43-0	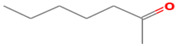	3 ^ad^	fragrance	**+**	**+**
1-penten-3-one	1629-58-9	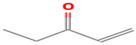	0.0009 ^ad^	spicy, onion	−	−
1-octen-3-one	4312-99	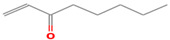	0.000003 ^a^	green beany	−	**+**
2-octanone	111-13-7	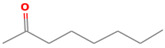	0.05 ^a^	soapy, floral	**+**	−
**Furan**						
2-penthyfuran	3777-69-3	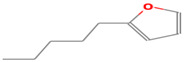	0.0058 ^ab^	beany	−	**+**
2-ethylfuran	208-16-0	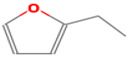	0.0023 ^b^	beany, earthy,	−	−
**Alcohol**						
1-octen-3-ol	3391-86-4	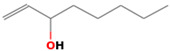	0.007 ^ac^	mushroom	**+**	**+**
1-pentanol	71-41-0	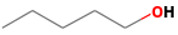	0.1502 ^ab^	green, wax	**+**	**+**
1-penten-3-ol	616-25-1	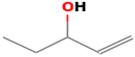	0.3581 ^ab^	beany, green	−	**+**
hexanol	111-27-3	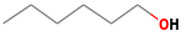	0.5 ^ab^	beany, green	**+**	**+**
3-methyl-1-butanol	123-51-3	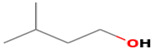	0.004 ^ab^	balsamic	**+**	**+**

## 3. Non-Volatile Taste Compounds

Aroma and taste metabolites have a close relationship. These metabolites, which are often non-volatile in nature, enhance the gustatory experience by accentuating the volatile aroma metabolites, which in turn contribute to flavor profiles. Umami, bitter, sour, salty, and sweet are the five categories into which taste experiences are categorised. Different chemical metabolites are involved in plant-based products taste perception [37]. Flavor perception of faba beans ingredients involves both interaction between volatile odorant compounds and non-volatile taste metabolites, which belong to different chemical classes. Hence, both volatile and non-volatile compounds influence the overall perceived flavor response in faba bean ingredients [32].

Bitterness in faba bean ingredients is often associated with alkaloids, polyphenols, tannins, peptides, and certain glycosides. Most of these metabolites are present in large concentration in raw faba bean [46,47]. The hull of coloured faba beans contains a total of 72 to 82% tannins. Proanthocyanidins also make up 96% of the tannins in seeds [48]. The primary phenolic substances found in faba include phenolic acids, flavonols, flavanols, flavanonols, flavanones, and isoflavones [49,50,51]. Polyphenols are secondary metabolites associated with bitterness, especially for the flavonoids class, although they have their associated health benefits related to anti-inflammatory function and other health benefits [52,53].

Other non-volatiles metabolites include hydroperoxy epoxides, hydroperoxy cyclic peroxides, and dihydroperoxides [32]. Saponins imparts bitter and metallic off-flavor taste to faba bean ingredients. However, saponins are usually heat stable and thus difficult to remove during processing, thus most saponins could be reduced mostly in concentrate and isolate. These non-volatiles compounds are usually bound to proteins and thus increase their stability. Soyasapogenol B (0.020 mg/g), soyasaponin Bb (0.040 mg/g), soyasaponin g, and azukisaponin IV are among the saponins found in faba beans [54,55,56]. Due to the inclusion of 2,3-dihydro-2,5-dihydroxy-6-methyl-4Hpyran-4-one moiety linked to its C22, soyasaponin g varies from the Bb type and imparts more bitterness [57]. Several techniques and technologies, such as debittering, have been employed to reduce the levels of these taste metabolites to increase consumer acceptance of plant-based products [52].

## 4. Effect of Post-Harvest Processing on Faba Bean Volatile Compounds

Different processing technologies are employed along the food chain to improve the flavor profile of pulses, such as thermal treatment, germination, debittering, fermentation, enzymatic treatment, masking, solvent extraction, pH shift, and ultrafiltration [18]. A major component that influences flavor profile is the nutritional profile, such as fat, protein, and carbohydrate content, due to specific binding interactions with volatile compounds.

Monitoring of volatiles along the faba bean food chain can be followed to investigate the impact of processing on aroma compounds with the occurrence of new targeted compounds. Since the evolution of these volatile compounds over time is poorly documented, some hypotheses have been suggested to explain variability in volatile profiles, such as transformation of volatiles into non-volatile compounds and vice versa. However, other phenomena can possibly be at work. For instance, the chemicals may be retained by the packaging, react with one another, or interact with the food matrix as it changes throughout processing to storage. The following section describes the current research on the impact of processing on the flavor profile of faba bean ingredients (Figure 1). 

### 4.1. Faba Bean Variety and Cultivation

Intrinsic aroma of pulses is greatly influenced by genetic makeup [18]. Faba bean is a legume-based protein that contains approximately 250 g protein/kg seed. There are over 43,000 faba bean accessions available in the GenBank database worldwide [6]. With regards to the large accessions of faba bean varieties, the volatile profile of only a few cultivars have been investigated. As the cultivar and species volatile profile varies from year to year due to climate conditions and cultivation conditions, this makes it difficult to make comparison among the same species.

The volatile profile of the faba bean cultivars was characterised [27]. Colored flower genotypes include Divine, Fatima, Florent, Melodie, SSNS-1, and Taboar, whereas the genotypes AO1155, Disco, Divine, and Melodie are low tannin and vicine-convicine varieties. Observed differences in the quantity of extracted volatiles was observed among the genotypes studied, For instance, Barrhead showed low quantity of extracted volatiles (FB25-56 and AZ10) and high abundance in NPZ4, Taboar, Fatima, Melodie, Disco, and Florent varieties. Analysis of variance showed that geographical location and genotype influence the volatile profiles of the studied faba bean genotypes. Aroma compounds, such as 2-ethylfuran and 2-pentylfuran, acetone, 2-heptanone, and 1-hexanol, were the main contributors in genotypic variation, while compounds, such as aromatic hydrocarbons, alkanes and furans, contributed to differences in environmental conditions. Flavor analysis using SMPME–GC–MS, together with multivariate analysis, was utilised as a screening strategy to find genotypes with preferred flavor characteristics.

Rajhi et al. [25] observed differences in volatile compound profile among four cultivars that were analysed. For instance, the vicia faba major variety was different compared to the vicia faba minor varieties. Previous research on the volatile in faba bean flour also revealed that volatile chemicals vary not only across variations with different tannin contents (low-tannin and high tannin), but also within a single type of variety [31]. Therefore, choosing or selecting a low hexanal genotype, such AO1155, may lessen the flavor profile’s unpleasant aroma compounds and boost customer acceptance of various products using faba beans. Investigation of the flavor profile of a wide range of faba bean genotypes is useful in selecting appropriate faba bean genotypes for faba bean breeders as part of quality attributes to meet market demands and acceptability.

Akkad et al. [44] investigated the volatile profile in high and low tannin Canadian grown faba beans. According to the Ehrlich-Neubauer Pathway, major groups of volatile compounds derived from the biodegradation of amino acids were observed, including aldehydes, alcohols, and acids that could be derived from leucine (3-methylbutanal, 3-methylbutanol, 2-methylbutanoic acid), isoleucine (2-methylbutanal, 2-methylbutanol, 2-methylbutanoic acid), and alanine (ethanol, acetic acid) (Spinnler, 2012). 

### 4.2. Dehulling

The hull of faba bean seeds constitute a substantial mass of the overall mass, hence, it has a strong impact on aroma profile. Dehulling of faba bean seed showed a significant difference compared to whole seeds. For example, volatiles of aldehydes, ketones, and alkanes were more in dehulled seeds compared to whole seeds. Following dehulling, it is expected that the seeds will be exposed to atmospheric oxygen that can initiate oxidation of oleic and linoleic acid. Thus, it is expected that dehulled and milled flours will tend to have higher levels of aldehydes than fresh whole seed. Compounds, such as alcohols and organic acids, were lower in hulled faba beans in comparison to whole seeds. However, 1-hexanol was twice as abundant in whole seeds as compared to dehulled seeds [44]. A detailed volatile profiling of faba bean shells needs to be investigated, as there is a paucity of information on them.

### 4.3. Substrate Composition

Yang et al. [58] investigated the role of lipid-modifying enzymes in the synthesis of volatile lipid compounds in faba beans, rapeseed oil, and rapeseed fatty acids as model food systems. Lipase and LOX activity were investigated at different concentrations and three different pHs. Faba bean extract alone, without addition of rapeseed oil (RO), was observed to be dominated by hexanal, pentylfuran,1-hexanol, and 1-octen-3-ol at all three pHs. Additionally, there was a notable variation in the quantity of key compounds at the three distinct pHs. The number of volatile chemicals significantly increased in the faba bean extract (3% RO and 5% RO) model food system. Only two volatiles, 2,4-heptadienal and 2-pentenal, were discovered to be present in emulsions, but not in extracts. At pH 6.4, the addition of 3% RO doubled, and the addition of 5% RO tripled the levels of volatile products compared to their respective extracts. The addition of RO did not significantly increase the quantities of volatile products at pHs 5 and 8. When compared to rapeseed oil emulsions, food models, using ROFA emulsions (3% and 5% ROFA), produced more volatile compounds. Variation in lipase activity in faba was observed within cultivation years, indicating that variation was predominantly based on differences among cultivars.

Numerous studies have investigated the influence of lipase in oats [59,60,61], but limited information is available regarding the role of lipase enzymes in faba beans. A previous study on lipase activity from small faba beans used various *p*-nitrophenyl fatty acyl esters [62], and another examined the activity and characterization of lipase in faba bean [63]. According to Yang et al. [58], faba bean lipase showed high affinity in hydrolysing short-chain fatty acid esters compared to long-chain FAs. Additional optimal pH testing with the aid of p-NPB revealed that the lipase was more active at pH 8.0 than at lower pH levels, and that no activity was seen below pH 6. According to earlier research, faba bean lipase was most active at an alkaline pH of 8.5 when employing the substrates tributyrin or olive oil [63] and *p*-nitrophenyl laurate [62] as substrates.

Faba beans from different cultivars have been found to have a high LOX activity [58,64,65]. These investigations found that faba beans have a moderate degree of LOX activity in comparison to other legumes, such as soybeans and lentils. Additionally, the faba bean samples’ considerably varying LOX activity levels showed that cultivars and growth conditions had a major impact on faba bean LOX activity. The highest optimum pH for LOX activity was also found to be pH 6 [58,66].

In conclusion, despite the low lipid content in faba beans, lipids should be given careful attention, since low levels of secondary lipid oxidation products may result in noticeable drawbacks [44], and, when oil is incorporated during product development, for example, in emulsions, a risk of enzymatic lipid degradation may occur. The high pH at which faba bean lipase and LOX function best also indicate careful consideration, especially when processing in mildly alkaline conditions to minimize production of undesirable off-flavors in food products. Factors, such as water activity and kinetics of lipase activity, depend on moisture content. Hence, controlling moisture content may limit enzyme activity. For instance, in wheat bran, lipase activity peaked at around 20% moisture content, while it was comparatively low at extremely low moisture concentrations (<10%) [67]. Studies on the impact of water activity on faba bean lipase activity in dry food matrices are currently limited. Hence, substrate composition is a crucial factor affecting volatile profile and sensory properties.

### 4.4. Milling/Grinding

Milling is a major processing step carried out during the production of faba bean flour. Friction occurring during milling leads to elevated temperatures that may lead to heat-induced degradation of flavor precursors, such as amino acid flavors. The impact of milling on odour alterations has been reported to influence volatile profile in pulses [68]. Ref. [25] analysed the volatile profile of whole seeds and flours of legume cultivars. A total of 35 volatile compounds were observed in whole seeds and corresponding flours. The predominate chemical group were monoterpenes H (Monoterpene Hydrocarbons) (from 3.4 to 36.9%), SH (from 0.6 to 89.1%) and non-terpene derivatives (from 5.9 to 65.0%). For most cultivars of faba bean, except VF-Bachar (faba bean minor), there was an observed reduction in the levels of monoterpenes after grinding into flour.

Rajhi et al. [25] noted that, after whole seeds were crushed into flour, sesquiterpene hydrocarbons and non-terpene derivatives increased (89.1 and 5.9%, respectively) in comparison to whole seeds (0.6 and 61.4%, respectively). For example, VF-Mamdouh revealed an inverse interplay between the whole and crushed seeds in terms of these compounds. In general, there was a reduction in the total number of aroma compounds from 28 to 16 after crushing into flour. Compounds, such as simphiperfol-5-ene and 7-epi-simphiperfol-5-ene were formed, as well as 1-nonanol and 2-undecanone. Heptanal, phenol, butyl butyrate, acetophenone, and other volatiles were among the compounds that were eliminated. Grinding ensures particle size reduction in order to increase the availability of both non-volatile and volatile compounds, as well as to increase surface area during extraction. Ref. [44] observed that grinding greatly increased the concentration of 2-butanone, 3-methyl butanoic acid, 1-octen-3-ol, and 2-pentylfuran in freshly milled faba beans. This suggests that flavor properties are influenced by milling conditions and thus need to be controlled during processing.

### 4.5. Thermal Processing

The influence of thermal treatment on faba beans is commonly used to reduce antinutrients, improve functionality, and reduce the activity of enzymes, such as LOX, that cause beany flavor development. Additionally, the Maillard reaction product is often employed to improve food flavor through heating. Ref. [69] reported that thermal treatment of faba bean flour caused a distinct change in peak area of aldehydes in conventional oven treatment and microwave treatment. Hexanal content was relatively higher in microwave treatment compared to conventional oven treatment, while non-treated faba bean had showed a lower mean area. In addition, nonanal level was higher in microwave treatment compared to conventional heat treatment and non-treated faba beans. This indicates the impact of thermal treatment on chemical oxidation and production of more off-flavor compounds. However, after suspension of faba bean flours into water, new volatile compounds were observed, which may indicate that certain enzymes were activated, and a more pronounced effect was seen in untreated faba beans. Alcohols, such as 1-penten-3-ol, octen-3-ol, nonanal, and furans, which are known to be generated by enzymatic processes, were among the odour-producing molecules that were identified (Kroft et al., 1993). This indicates that, depending on the solvent in which the flour is suspended, it greatly influences the release of aroma compound. Preparing faba bean flour in oil emulsions was also characterized by more volatile compounds. Particularly, untreated emulsion showed higher volatiles (mostly for aldehydes and alcohols such as hexanal, nonanal, and hexanol) compared to microwave treatment and conventional treated faba beans flours. According to various food matrices, hexanal has been linked to various sensory notes, such as green and leafy perception.

Despite the promising nutritional, health, and environmental benefits of faba beans, there has been limited research on the impact of thermal processing on sensory properties. Ref. [69] examined the association between the sensory qualities of thermally treated faba bean samples and volatile profiles. Thermal treated faba bean samples (conventional oven and microwave heating) were mostly attributed to pea and beany odour perception. Particularly, oven-treated flour was associated with nutty and grain impression notes, while microwave treatment was judged to have a fermented, grassy, and rancid odur. In addition, fermented and grassy odor was found to due to the presence of 1-penten-3-one. Aldehydes, such as 2-octenal, were also attributed to cause beany notes.

The demand for plant-based ingredients is rapidly becoming prevalent, and there is a high market for plant derived beverages. Nevertheless, commercially developed products are faced with challenges in terms of nutritional profile, functionality, and off-flavors. As most emulsions are unstable, plant proteins can adsorb at the oil–water interface and thus stabilise the emulsion. Ref. [70] studied the volatile profile of model faba bean-based beverages at each stage of processing and manufacturing. A total of 21 compounds were observed and quantified. An observed incremental increase in the concentration of volatiles was observed at each step in the production of the model beverage (homogenization and ultra-high temperature treatment 105 °C for 3 s and heat treatment at 135 °C for 3 s). Most volatiles were generated and increased from lipid precursors in faba bean concentrate due to lipid oxidation because of thermal treatments on lipid components. Pentanal, hexanal, heptanal, E-2-heptenal, and 2-pentyl furan were among the odorants that were produced, and they were generally linked to the oxidation of linoleic acid, the most predominant fatty acid in faba beans and sunflower oils. Nonanal may have been generated through oxidation of oleic acid. However, following refrigeration of model faba bean beverages for four weeks, there was a reduction in the concentration of volatiles, such as hexanal and pentanal, with concomitant increment in corresponding alcohols, such as 1-pentanol and 1-hexanol, which may be due to enzymatic and/or microbial degradation.

In both wet and dry fractionation for faba bean protein enrichment, the oxidizing agent present in the raw material usually appears after protein extraction. Hence, the beany flavor becomes a nuisance and is transferred throughout the pre-processing step, so these enzymes must be inactivated. Processing techniques, such as autoclaving, conventional heating, steaming, and microwave treatment, have been recommended to remove and/or mitigate beany flavor through the inactivation of enzymes [71,72]. Additionally, pre-treatments, such as dehulling and blanching, prior to milling to avoid lysis of cells that may cause contact between enzymes and lipid molecules, were performed.

Jiang et al. [73] investigated the impact of thermal treatment on faba bean seeds using a conventional oven and microwave heating at different times. Microwaving heating (950 W) at longer duration (>2 min) and conventional heating at 170 °C for 30 min reduced lipoxygenase and peroxidase to undetectable levels. The inactivation of these enzymes, which act as precursors for lipid degradation, indicate the effectiveness of thermal treatment. Changes in nine volatile compounds were observed following thermal treatment. Unheated faba bean seed was strongly associated with 2-methylfuran, an indicator of enzyme activity, as well as 2-pentylfuran. However, thermally treated samples showed low 2-methylfuran and 2-pentylfuran, indicating the effectiveness of inhibiting or inactivating enzymatic activities. Volatile compounds, such as hexanal and nonanal, were typically high in conventionally treated faba bean seeds and microwave-heated samples for two minutes. High amounts of hexanal and nonanal for thermal treatment clearly indicate lipid autoxidation products following exposure of seeds to elevated temperature or duration. Hence, microwave heating for 1.5 min represents a good compromise to avoid lipid autoxidation while inactivating enzymatic activity.

Akkad et al. [31] also exposed sprouted faba bean flours to different temperatures to investigate the impact of drying conditions on volatile profile. Drying times of 24, 36, 48, and 60 h were investigated in sprouted faba beans. Different drying temperatures were found to affect volatile peaks. Drying for 24 h was found to be most desirable, as higher duration resulted in an increased level of compounds, such as aldehydes. Selecting appropriate drying temperature and time is crucial in preparing seeds for further treatment, such as milling and protein extraction. In conclusion, drying at appropriate drying temperature and time need to be assessed during the processing of faba bean ingredients.

### 4.6. PH

PH can have a significant impact on flavor due to the complex interaction between volatile compounds and protein molecules. Protein binding flavor compounds are influenced by the structural properties of the protein and the nature (length and functional group) of flavor compound [74] and extrinsic conditions, such as pH. Interaction between protein and aroma compounds could be reversible or irreversible, depending on specific conditions that affect hydrophobic interactions and hydrogen bonds [75]. Reversible interactions are beneficial for preserving favourable flavor components after processing and throughout consumption [76].

Sharan et al. [45] explored the volatile profile of faba bean concentrate suspensions and observed eighty-three volatile compounds for pH_utilization_ 4 and 7, which were comprised mainly of alkanes, alkenes, alcohols, aldehydes, ketones, esters, organic acids, aromatic hydrocarbons, furanoids, and pyranoids. Utilization of faba bean concentrate at pH 4 resulted in a higher level of aldehydes, representing 76% of the total chromatogram compared to pH_utilization_ at 7, which showed only 10%. However, pH_utilization_ at 7 also showed higher signals for alcohols (72%), while, at pH_utilization_ 4, the total peak area of alcohol represented was 13%. Utilization of FBIC at pH 4 generated 26 more volatile compounds compared to pH_utilization_ at 7. These compounds included aldehydes, alcohols, furanoids, one alkane (tetradecane), one alkene (1-heptene), ketones (2-heptanone, 3-octen-2-one) and organic acids (octanoic acid, nonanoic acid). This indicates that pH plays an important role in influencing volatile profile of faba bean ingredients, which may improve or be detrimental to the flavor of end-products. FBC was also subject to modification by the pHprocess. Variation in volatile profile was monitored at different processing pHs of 2, 4, 6, and 11, which produced volatile compounds of 86, 87, 72, and 78, respectively. Observed difference in volatile levels was observed for all aroma compounds, except furfural, 3-methylbutylacetate, 2-ethyl-1-hexanol, and 9-octadecanal when FBC was subjected to different processing by pHprocess, Temperature_process_, and time_process_. However, generally, it was observed that the pH modification process generated higher volatiles compared to non-pH modified FBC. This suggested that process modification with pH may have influenced alterations in flavor through different reactions. Additionally, the influence of the temperature process and the time process were crucial in determining the likelihood that these volatile-generating reactions would take place.

During extraction of protein from faba bean flours, the extraction method investigated whether alkaline/isoelectric extraction, ultrafiltration, salt extraction, and micellar precipitation will greatly influence the composition of concentrates or isolates obtained and modify aroma molecules that may be present. There is very few information on aroma compounds present in faba bean concentrates or isolates, and, thus, this requires a thorough investigation. New flavor compound evolution may occur during each step of the extraction process due to release of volatile from binding sites of lipids and proteins and provide useful information in generating desired final products.

In conclusion, there were two distinct effects: first, the impact of pHutilization, which was immediately noticeable for the FBC and less pronounced, but still significant for all the modified ingredients, and, second, the effect of process modification with pH, which was noticeable when examining the relative proportions of the various volatiles.

### 4.7. Fermentation

Fermentation represents a promising mean of reducing the levels of off-flavors in faba beans and generating new desirable flavor compounds. Starter cultures are mostly applied in the food industry because they possess specific enzymatic activities to modify substrates. The presence of high amounts of protein in pulses, such as faba beans, comes along with challenges, such as susceptibility to degradation, which may generate off-flavors or bitter-tasting peptides. Mitigation of fatty acid oxidation could be accomplished by focusing on precursors of oxidation and controlling processing conditions.

Firstly, defatting strongly reduces the presence of precursor oils and results in substantial reduction in off-flavors prior to processing [77]. Additionally, reduction in off-flavor may be accomplished using fermentation. For instance, during fermentation, volatile compounds, such as hexanal, may be converted into hexanol [78] or hexanoic acid [79], with less perception of being off-note [80]. Hence, fermentation is a promising means to eliminate and/or reduce undesirable aroma compounds and is widely applied in the food industry. Lactic acid fermentation may contribute two different benefits to sensorial properties of faba bean ingredients. Principally, this is performed by reducing beany flavors below their odour detection limit and, secondly, by generating new odour compounds, which can mitigate beany flavors or mask them to produce an overall improved aroma perception. By modulating the legume-based substrate, to favour the availability of nutrients for the microorganism and decrease antinutrients, the subsequent lactic acid fermentation can be aimed to improve aroma and taste attributes. In addition, fermentation could contribute more benefits by reducing levels of antinutrients, such as tannins and raffinose fructose oligosaccharides, which cause gastric disorder.

Improvement in overall aroma has been documented for lupine [80], soy [81], and pea [82]. A decrease was observed in hexanol, and an increase in hexanoic acid was observed with a mix of *L. paracasei* and *L. rhamnosus* [83]. Increased concentration of hexanoic acid was also observed for *L. plantarum* and *S. thermophilus* in soy [81]. Fermentation of pea protein with mixed lactic acid bacteria culture resulted in a significant reduction in hexanal content and an increase in hexanol levels [82].

Apart from the selection of an appropriate strain, variation in fermentation parameters has drastic impact on the aroma profile produced by lactic acid bacteria. In conclusion, lactic acid fermentation has been successfully utilized to improve the aroma of pulse-based ingredients. Selection of appropriate strain and controlled fermentation parameters are needed to produce desirable aroma compounds. More research on fermentation to improve aroma profile of faba bean ingredients is limited. Thus, more scientific work is needed to fill this gap.

### 4.8. Storage

Akkad et al. [24] analysed the impact of different storage conditions on the volatile profile of Canadian faba bean flour. Following one week of storage, no observable difference was observed in aldehyde content for all the conditions. However, after two weeks of storage at room temperature and refrigeration, there was a significant increase in aldehyde content. Storage of faba bean flour for two months resulted in an increased aldehyde content by about 80% and 28% under room temperature and refrigeration, respectively. In the case of flour stored in the freezer, there was a slight increase after two weeks and then decrease. Under all conditions, alcohol reduced dramatically; ketones also saw alterations, albeit to a far lesser extent. During low-temperature storage, the presence of D-limonene considerably boosted the overall abundance of alkenes. In comparison to samples taken at ambient temperature, the headspace of samples that had been chilled and frozen contained considerably more aromatic hydrocarbons. Toluene and ethylbenzene, which were both present in this class and increased with storage duration under all storage circumstances, dominated.

According to Jiang et al. [73], the main volatiles responsible for faba beans’ beany flavour include 2-methylfuran, 2-pentylfuran, hexanal, nonanal, and 2-heptanone, which are produced by oxidation of linoleic and linolenic acids. Akkad et al., 2021, examined the relative odour activity values (ROAV) of faba bean flours stored at different conditions in order to identify key aroma compounds. These compounds include hexanal, octanal, nonanal, decanal, (E)-2-nonenal, 1-hexanol, 1-octen-3-ol, and octanoic acid, originating from unsaturated fatty acids; 3-methylbutanal, phenyl acetaldehyde, phenyl ethyl alcohol, acetic acid, and 3-methylbutyric acid were derived from amino acid reactions and degradation products, as well as volatiles from other origins, including b-linalool. Formation of beany flavours was more pronounced during storage at room temperature and affected by storage time compared to refrigerated storage or frozen storage. Similar observation was made by [44] during storage of milled low and high tannin faba bean flours. A significant difference was observed for aldehydes, organic acid, esters, and alkanes. Lipid oxidation products, such as 1-hexanol, nonanal, and hexanal generated from linoleic and oleic acid, dominated more in stored samples than freshly milled faba beans. This indicates that, despite the low lipid content of faba beans, storing for prolonged duration could have a detrimental effect on its flavour profile.

In conclusion, the storage of faba bean flour at room temperature and refrigerated conditions can last for a maximum of two weeks without any serious significant impact on volatiles and specific beany flavours. However, for a longer duration of storage, freezing of flour is mostly recommended to inhibit enzymatic conditions that may result in lipid degradation. Further studies in this area are needed to support these findings, as there is limited information in this area.

### 4.9. Germination

Germination is a widely used technology in food processing due to its considerable influence on textural properties and flavor over un-germinated samples. During germination, plant enzymes, such as lipases, amylases, and proteases, are released, resulting in the breakdown of lipids, as well as polymers, such as carbohydrates and proteins, respectively. While sprouting is beneficial for flavor improvement, lipase degradation can cause a counteractive effect through autoxidation by the activity of lipase, thus generating off-flavors [84]. Apart from flavor enhancement that results from sprouting, reduction in the levels of antinutrients, such as tannin, has also been documented [85]. A reduction in tannin content is beneficial, as this improves absorption of proteins and minerals, such as iron and zinc. In addition, tannin is implicated in contributing to bitter perception. Hence, sprouting may resolve some of these sensory drawbacks in pulses. Ref. [31] investigated the impact of sprouting on flavor quality of faba bean flours. Following germination for 24, 48, 52, and 74 h and subsequent drying at 60 °C, germination for 48 h showed the least amount of absolute abundance. For all germinated samples, alcohol and ketones were the most abundant. Beany flavor compounds, such as hexanal and nonanal, showed the lowest abundance after 48 h. Most aroma-active compounds (20 out of 33) considerably reduced after germination, whereas 10 compounds rose in abundance. However, 10 pleasant aroma compounds were found to increase after germination. Volatiles originating from amino acid degradation were also observed to increase after germination. In prior research comparing the amino acid profiles of germinated and ungerminated faba bean flours, it was shown that sprouting dramatically reduced the amounts of isoleucine and leucin [84]. It is well known that the Strecker degradation or enzymatic mechanisms can cause the breakdown of isoleucine and leucine to produce the aldehydes 3-methylbutanal and 2-methylbutanal. High sprouting duration over 48 h was observed to have an adverse effect on beany aroma compounds. In conclusion, high sprouting duration over 48 h was observed to have an adverse effect on beany aroma compounds.

## 5. Other Flavor Modifying Processing Techniques

Different emerging processing technologies to improve flavor profile of pulses are gradually receiving more attention. Among them include chemical processing, which involves incorporation of metal-reducing agents or dehydrogenase to transform aldehydes into their respective acids [86,87]. Chemical methods for removing or reducing flavor compounds possess limitations regarding residual solvent, which may raise safety concerns. Enzymatic treatment using Protamex enzyme (pH 6.0) at 45 °C for 40 min has been shown to reduce the levels of beany flavour by 70% in soy protein isolate [88]. Genetic engineering to produce faba bean cultivars lacking in LOX or with lower concentration of LOX may play a significant role in reducing the levels of undesirable flavor compounds in faba beans.

*β*-Cyclodextrin has also been documented in various studies to improve flavor profile [89]. *β*-Cyclodextrin is a cyclic oligosaccharide capable of entrapping and removing flavor compounds due to its highly hydrophilic outer surface and hydrophobic cavity [90]. Very little information is available on the impact of texturization on the flavor of faba bean products. During texturization, maillard reaction products, such as pyrazines and furans, increase, while volatiles, such as alcohols, ketones, acids, and esters, are drastically reduced [91]. This processing technique is a useful technique to generate varying textured products with different odour-active compounds with improved functionality.

In conclusion, innovative and emerging processing methods include microwave treatment, electric field heating, supercritical fluid extraction, ultrasound-assisted extraction, plasma technology, addition to flavor compounds and treatment with antioxidants. These novel innovative technologies present numerous advantages in improving flavor profile. However, impact on functionalities needs to be investigated alongside their influence on flavor compounds. Consideration in terms of environmental sustainability and nutritional profile also needs to be evaluated, as well as safety with regards to these innovative methods.

## 6. Impact of Processing on Sensorial Properties and Consumer Acceptance

Although there is limited information on the interplay between process conditions, odour perception, and volatile chemistry, attempts have been made to better understand this interaction in flavor development and perception. Different processing techniques, such as cold soaking of faba beans, followed by thermal treatment by microwaving and hot extraction of faba bean milk, were used to produce faba bean yoghurt with the aim of reducing beany flavor in the final product. However, faba bean yoghurt generated a higher rating of beany flavor following the consumer acceptability test [92]. Processing FBC at pH 4 and 6.4 was linked to the perception of more milky, fresh, and fruity blossom attributes. In the case of FBC suspension at pH process 6.4, aldehydes and alcohols were the most prevalent, with few differences regarding the extent of processing (Temperature_process_ and time_process_). In the majority of the constituent suspensions at pH4 series, aldehyde and terpenoids signals predominated [45]. Terpenoids, with a few alcohols and aldehydes, are frequently associated in relation to fruit aroma [93].

A higher degree of processing of FBC resulted in perception of a more cooked odour for pH 2 (winey, meaty, and ammoniac notes) and pH 11 (smoky and burnt aroma attributes). High volatile complexity was also linked to ingredient change at pH process 2. Previous studies have demonstrated that acidic processes in protein-rich systems result in protein hydrolysis, where fewer hydrophobic interaction lessen the possibility of flavor binding and permit more flavor release [94,95]. However, in protein-rich matrices, alkaline processes allow for increased protein binding to molecules, such as butanal, hexanal, and 2,3-butanedione, as well as increased ketone release, including 2-heptanone [96]. Modification of faba bean concentrate at pH 11 showed considerably larger signals for ketones and aldehydes and may have been affected by this matrix effect. These ketones come from the oxidation of lipids, including 3-octen-2-one (from linoleic acid), 3,5-octadien-2-one (from -linolenic acid), 2-butanone, 2-pentanone, and 2-heptanone (from linoleic and -linolenic). Additionally, the thermal breakdown of sugars and amino acids, such as alanine, valine, isoleucine, and cysteine, yields 2-butanone and 2-pentanone (e.g., glucose, sucrose) [44]. Deamidation, which occurs when proteins are processed in an acidic or alkaline environment, releases ammonia at high temperatures as a result of the conversion of asparagine and glutamine residues into their carboxylic forms. This explains why, in some cases, a high amount of ammoniac odour compounds was generated [97].

### Applications of Faba Beans in Product Development

Owing to their nutritive value and techno-functional properties, faba beans are gaining wide application in a variety of food products, such as sausages, as well as meat analogs as alternatives to traditional ingredients, such as casein, whey, and wheat protein. This section reviews the potential of faba beans in food-related applications (Table 2). The high phenolic content and antioxidant activity indicate the potential for incorporating faba bean flours into different formulations to increase the levels of bioactive compounds. Ref. [51] investigated the nutrient profile of wheat–faba bean composite flour pasta (35% of faba bean flour).

Faba bean composite flour revealed a 190 and 18% increase in phenolic content and antioxidant activity, respectively. A similar study by [98] developed unleavened crackers from wheat–faba bean composite flour. There was an observed increase of 162% in total phenols and 182% in antioxidant activity after faba bean substitution at 35%. Fermented faba bean also represents an interesting technology to develop food products that confer specific nutritional attributes. Lactic acid bacteria used in fermentation can increase the quantity of free and essential amino acids and produce γ-aminobutyric acid (GABA) and also enhance protein digestibility [99].

Recently, protein isolates from pulses are not only used for their nutritional benefits, but they are also incorporated to provide specific health benefits. A low amount of antinutritional factors and bioactive properties of protein isolate allow substitution of food products to confer additional health benefits. For instance, some studies have incorporated protein isolate and concentrate from faba bean to produce mayonnaise and meat analogs. Faba bean isolates added to mayonnaise at 3% showed antioxidant activity, ACE inhibitory activity, as well as a-glucosidase inhibitory activity [100], indicating the health benefits of using isolate. More importantly, the fabricated mayonnaise showed lightness and redness, which may be superior to that of the control. It will be interesting to further study the organoleptic properties of the fabricated mayonnaise and feeding trials to validate the stated health benefits.

**Table 2 foods-12-02237-t002:** Edible products developed from faba bean ingredients.

Sample Type	Application and Major Results	References
Flour	Faba bean flour up to 20% *w*/*w* can be incorporated into sausages.	[101]
Flour	Replacing a portion of wheat flour with 30% fermented flour resulted in pasta with a homogenous texture and lower cooking loss compared to a 50% substitution. Pasta produced from unfermented faba bean flour resulted in an improvement in nutritional value without any impact on sensory and technological features.	[99]
Flour	In comparison to bread made using soybean flour, a 50% substitution with fermented and unfermented faba bean flour produced a bigger loaf volume. Partial replacement with fermented faba bean flour did not affect sensory score.	[102]
Flour	Tofu and yogurt produced from whole faba bean flour, using two different methods, showed typical emulsion properties.	[103]
Flour	Incorporation of faba bean flour at 40% showed increased protein content and antioxidant activity. Faba bean crackers were more acceptable compared to control wheat flour.	[98]
Flour	Tofu-like curd was produced using faba beans, which were found to be high in protein content and high in in vitro protein digestibility.	[104]
Flour	Incorporation of faba bean flour showed a rise in water absorption and reduction in dough development time as substitution level increases. Up to 30% substitution increased nutritional characteristics without impacting flavor and the physiochemical properties of faba bean-formulated pasta.	[105]
Flour	Faba bean flour used to enrich semolina flour at different concentrations resulted in a lower cooking time and higher dry matter loss; enhanced nutrient content and increased micro and microelement content were observed in enriched pasta.	[106]
Flour	Gluten-free pasta produced from fermented and unfermented faba bean flour resulted in higher cooking loss and lower water absorption compared to semolina flour. However, fermentation adversely affected textural and flavor attributes. Transglutaminase-treated pasta produced from faba bean flours decreased in vitro starch hydrolysis index and impacted some textural properties.	[107]
Flour	Up to 80% replacement with faba bean flour in pasta resulted in a sensory score close to commercial whole wheat pasta. Increasing the percentage of faba bean flour from 0 to 100% resulted in an increased cooking loss, decreased resilience, and reduced covalent network structures in the pasta.	[108]
Flour	Faba bean flour was to produce extrudates, which showed a higher expansion ratio compared to other beans.	[109]
Flour	Gluten-free pasta produced from faba beans was rich in protein, dietary fibre, and reduced antinutritional factors and glycaemic index.	[110]
Flour	Tofu was fabricated using faba bean, which showed a very pale colour. Enzymatic treated faba bean flour generated tofu with a similar texture and firmness as that of soya tofu.	[111]
Flour	Addition of faba bean flour at 30% in a wheat-bread formulation resulted in increased nutritional profile, especially protein content, with no significant impact on technological properties.	[112]
Husk	Faba bean husk-enriched bread at 33% was more satiating compared to plain bread. However, consumer acceptability was slightly lower	[113]
Flour	Using faba bean flour to produce tortillas and pitas resulted in a dark-colored product with beany flavor and bitterness.	[114]
Flour	Adding faba bean flour to durum wheat semolina at 35% increased the hardness and lowered the cooking loss of the pasta.	[115]
Flour	Faba bean flour supplemented into porridge increased protein content.	[116]
Flour	Partial or full replacement of wheat flour by faba bean increased the hardness, while flavor and taste were affected.	[117]
Concentrate	This had the potential to produce a meat-like product using faba bean concentrate with varying flavor and texture properties.	[15]
Concentrate	Faba bean concentrate was used to produce meat analogs using high-moisture extrusion. Meat analogues from FBC produced with temperature ranges from 130 to 140 °C showed the most preferred sensory and textural features, similar to the product category.	[15]
Concentrate	Faba bean concentrate can be used to develop infant formulas with similar in vitro protein digestibility as whey protein.	[118]
Concentrate, isolate	Faba bean ingredients (concentrate and isolate) can replace egg-based mayonnaise with no adverse impact on texture and sensorial properties.	[17]
Concentrate, isolate	High moisture meat analogs were produced using faba bean concentrate and isolate.	[119]
Concentrate, Starch concentrate, Fiber	Faba bean ingredient were used successfully to produce protein, starch, and fiber-rich foods	[16]
Protein Micellar Mass	Faba bean PPM incorporation into a novel snack and meat analogue showed good acceptability.	[120]
Hydrolysate	There were no appreciable changes between apple juice and the majority of the other hydrolysates evaluated when faba bean hydrolysates were added at 1% (*w*/*v*) to achieve fortification.	[121]
Isolate	Nutritional formulated products with FBPI contained VC at 13 mg/kg. Therefore, the equivalent concentrations provided by one serving (11 fl. oz.) of the NP would be around 5 mg of VC.	[122]
Isolate	Incorporation of 3% *w*/*w* FBPI to formulate mayonnaise resulted in enhanced antioxidant activity, as well as antihypertensive and antidiabetic activity.	[100]
Isolate	Mayonnaise produced using FBI at 0.375 and 0.5% could substitute conventional formulation.	[123]
Isolate	Biodegradable films were produced from faba bean isolate with enhanced mechanical and barrier properties.	[124]
Isolate	Incorporation of 20% *w*/*w* FBPI to beef mince resulted in improved nutrient content.	[125]
Flour, starch concentrate, protein concentrate and isolate	Substitution of 25% semolina wheat flour with faba bean ingredients (flour, starch concentrate, protein concentrate, and isolate) for preparation of pasta resulted in reduced postprandial glycaemia and appetite. However, protein content and pasta quality were greatly improved.	[113,126]
Flour, starch, concentrate, protein concentrate and isolate	Formulation of pasta using four faba bean ingredients flour, starch concentrate, protein concentrate, and isolate) at 25% replacement improved protein content and dietary fiber content.	[127]

Another major advantage of the utilization of protein isolates is due to their reduced concentration of antinutritional factors, which implies low allergenicity. This was observed in a study by [122], were the concentrations of vicine and tyramine were substantially low after incorporating into a formulated product. Similar reduction in antinutritional factors was observed by [108] during the production of pasta from legumes, such as faba beans. Thus, incorporation of faba beans ingredients may be an alternative solution to minimize the health effect caused by high concentrations of antinutrients, which affect nutrient bioavailability However, it is unclear whether reduction in antinutrients occurs during the processing of faba bean flour or the pasta production process.

Pitas produced using micronized and roasted faba bean flours caused a reduction in beany and bitter flavors. However, roasting generally received a high response for no beany and no bitter compared to micronizing [114]. Faba bean protein was used in the formulation of fat/cholesterol content in mayonnaise [123]. The emulsions produced from different concentrations of faba bean protein isolate (FBI) showed an excellent emulsion stability due to the ability of the proteins to adsorb readily at the oil–water interface and reduce the interfacial tension. Although fat is essential for sensorial properties, high concentration in products, such as mayonnaise, may result in emulsion instability and accompanying health implications. Substitution with faba bean protein could serve as an alternative emulsifier to egg yolk. Viscoelastic properties of faba bean protein emulsions and egg yolk were found to be similar, implying equal viscoelastic characteristics. However, textural parameters, such as firmness, cohesiveness, and adhesiveness, were found to be significantly affected after the incorporation of faba bean proteins.

Faba bean isolate was successfully used to develop edible packaging films reinforced with cellulose nanocrystals [124]. Fabrication of different biodegradable films using FBI mixed with cellulose nanocrystals (1, 3, and 7 wt%) and denatured by heating at 85 °C for 30 min in water was followed by cooling at room temperature and casting on a Petri dish. The films produced showed improved thermal properties and moisture barrier properties because of the interaction between cellulose nanocrystals and proteins. Further works exploring the impact of the films on sensory properties should be assessed, as well as the potential bioactive properties of the films. In addition, future works exploring using faba bean by-products (starch and oil) could be sustainable means of using faba bean ingredients after protein isolation.

## 7. Conclusions and Recommendation

Faba bean seeds are a multi-component system usually processed into different forms, such as flour, concentrate, or isolate; however, the majority of flavor analysis profile has been based on faba bean flour. The non-volatile and volatile compound significantly influence the perceived flavor qualities of faba bean ingredients, which may be present natively or formed during processing. However, several intrinsic and extrinsic aspects along the faba bean value chain can affect specific concentrations of these aroma compounds. In this review, we have discussed how the final faba bean product can be altered by different processing conditions, either separately or in combination. Despite the low lipid content of faba beans, their relatively high concentration of lipoxygenase leads to the formation of off-flavors, along with other reactions involving proteins, sugars, and polyphenol degradation. Effectively managing the development of off-flavors while avoiding the denaturation of faba bean proteins, preserving desired functionality, and maintaining nutritional value, are all extremely important. 

Germination, fermentation, and pH modulation were identified as promising methods to improve overall flavor and bitter compounds. Thermal treatment provides several advantages in modifying beany flavors of faba bean due to inactivation of lipoxygenase and lipase activity implicated in beany flavor formation. However, elevated temperatures required for inactivation of these enzymes may denature proteins and impair their functionalities. Emerging technologies, which are effective at short treatment time to inhibit these enzymes, represent a promising method to remove off-flavor while mitigating the impact of prolonged heating on functionalities. Thus, another important scientific advancement that needs attention is the implementation of non-thermal processing techniques and conditions due to their lower energy consumption, safety, reduced processing time, and potential to avoid the use of organic solvents. In this regard, ultrasound extraction, supercritical fluid extraction, Ohmic heating, and radiation techniques have been established and demonstrated to be effective in improving flavour characteristics. However, the scalability and economic viability in comparison with traditional methods remain a paramount hurdle with this emerging technique. Another promising area to improve and optimize flavor of faba beans is Maillard induced reaction. This represents a great research area that could improve off-flavor in faba beans. Additional, *β*-cyclodextrin ability to form complexes with numerous compounds may provide synergistic pathways to eliminate beany flavor in faba bean ingredients.

The role of enzymes in imparting flavor in faba bean ingredients needs further investigation during utilization conditions, especially for isolate and concentrates along the extraction chain, since there is limited research on flavor profile of faba bean protein extract. Hence, it remains unclear how to differentiate between the native profile of faba bean flours and protein extracts and at which point during protein extraction does fat oxidation and enzyme inactivation occur during the processing stages. For future research, breeding also provides a useful opportunity to reduce levels of polyunsaturated fatty acids or other precursors implicated in beany flavor formation.

## Figures and Tables

**Figure 1 foods-12-02237-f001:**
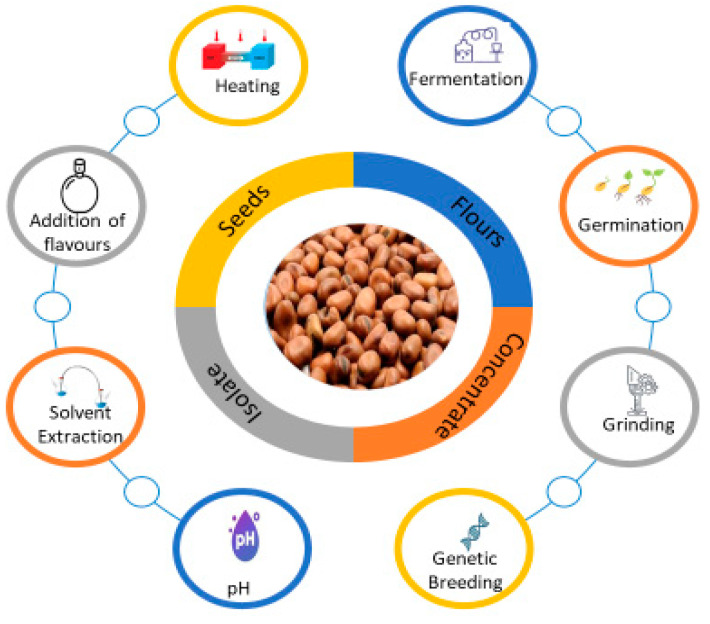
Processing treatment affects the aroma profile of faba beans ingredients.

## Data Availability

The data generated during the current study are available from the corresponding author upon reasonable request.

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
