# Peer review of "Faba Bean Flavor Effects from Processing to Consumer Acceptability"

_foods, 2023, doi:10.3390/foods12112237_

Round 1
Reviewer 1 Report
This review comprehensively provides up-to-date information on the volatile profile of faba bean ingredient and various processing techniques applied to modify flavor components. Volatiles and non-volatiles of faba bean ingredients have major impact on the perception of flavor attributes of faba bean final product.
The full text is written in detail, but the overall length of the article is relatively long. The summary of the various factors affecting the flavor during the processing of faba bean is not yet concise. It is suggested that the full text needs to be refined to clarify which issues related to faba bean flavor have been solved in existing research, and which aspects lack research. Therefore, further in-depth discussion is needed in the conclusion outlook.
Author Response
Response: The conclusion outlook has been modified to summarize existing research related to faba bean flavor and key areas that need further research.
Conclusion (Modified)
Faba bean seeds are a multi-component system usually processed into different forms such as flour, concentrate or isolate; however, the majority of flavor analysis profile are been based on faba bean flour. The non-volatile and volatile compound significantly influence the perceived flavor qualities of faba bean ingredients which may be present natively or formed during processing. However, several intrinsic and extrinsic aspect along the faba bean value chain can affect specific concentrations of these aroma compounds. In this review, we have discussed how the final faba bean product can be altered by different processing condition either separately or in combination. Despite the low lipid content of faba beans, their relatively high concentration of lipoxygenase lead to the formation of off-flavors along with other reactions involving protein, sugars and polyphenol degradation. Effectively managing the development of off-flavor while avoiding the denaturation of faba bean proteins, preserving desired functionality, and maintaining nutritional value are all extremely important.
Germination, fermentation, and pH modulation were identified as promising methods to improve overall flavor and bitter compounds. Thermal treatment provides several advantages in modifying beany flavors of faba bean due to inactivation of lipoxygenase and lipase activity implicated in beany flavor formation. However, elevated temperatures required for inactivation of these enzymes may denature proteins and impair their functionalities. Emerging technologies which are effective at short treatment time to inhibit these enzymes represent a promising method to remove off-flavor while mitigating the impact of prolonged heating on affecting functionalities. Thus, another important scientific advancement that needs attention is the implementation of non-thermal processing techniques and conditions due to their lower energy consumption, safe, reduced processing time, and potential to avoid the use of organic solvents. In this regard, ultrasound extraction, supercritical fluid extraction, ohmic heating, or radiation techniques have been established and demonstrated to be effective in improving flavour characteristics. However, the scalability and economic viability in comparison with traditional methods remain a paramount hurdle with these emerging technique. Another promising area to improve and optimize flavor of faba beans is Maillard induced reaction. This represents a great research area that could improve off-flavor in faba beans. Additional, β-cyclodextrin ability to form complexes with numerous compounds may provide synergistic pathways to eliminate beany flavor in faba bean ingredients.
The role of enzymes in imparting flavor in faba bean ingredients, needs further investigation during utilization conditions especially for isolate and concentrates along the extraction chain since there is limited research on flavor profile of faba bean protein extract Hence it remains unclear how to differentiate between native profile of faba bean flours and protein extracts and at which point in the protein extraction does fat oxidation and enzyme inactivation occurs during the processing stages. For future research breeding also provides a useful opportunity to reduce levels of polyunsaturated fatty acids or other precursors implicated in beany flavor formation.
Reviewer 2 Report
The authors have been done an excellent job. The article provides a combrehensive view in the field of Faba bean flavor. Though it is rather long, it covers virtually all aspects of flavour from a biochemical/chemical view throughout processing for various faba product/ingeredient development.
Save a few typing errors (line 492, 596, 612, 668) the manuscript is very well written.
Althought
Author Response
Comment: typing errors (line 492, 596, 612, 668)
Response: Typing errors listed above has been amended
Reviewer 3 Report
The manuscript was very interesting and it has enough references. On the other hand, it should be modified to give information clearly. The suggestion lists are ;
The abstract should be rewritten because it s not clear and does not contain all the stages of the manuscript.
The tables should be reformatted because they were too long and can t show clearly.
Section 7 can be enlarged.
Author Response
comment: (1) The abstract should be rewritten because it’s not clear and does not contain all the stages of the manuscript. (2) The tables should be reformatted because they were too long and cannot show clearly. (3) Section 7 can be enlarged.
Response: (1) The abstract has been modified to contain all stages discussed. (2) The tables have been reformatted to be clear. (3) Section 7 has been enlarged
Abstract
Faba beans as an alternative source of protein have received significant attentions from consumers and the food industry. Flavor represents a major driving force that hinders the utilization faba beans in various products due to perceived off-flavor. Off-flavor are generally produced from degradation of amino acids and unsaturated fatty acids during seed development and post-harvest processing stages (storage, dehulling, thermal treatment, and protein extraction). In this review, we discuss the current state of knowledge on the aroma of faba bean ingredients and various aspects such as cultivar, processing, and product formulation that influence flavour. Germination, fermentation, and pH modulation were identified as promising methods to improve overall flavor and bitter compounds. The probable pathway in controlling off-flavor evolution during processing has also been discussed to provide efficient strategies to limit their impact to encourage the use of faba bean ingredients in healthy food design.
Round 2
Reviewer 1 Report
This article provides a detailed discussion on Faba bean Flavor Effects from processing to consumer acceptance. The author has also made revisions to the abstract and conclusion outlook, and I believe there are some areas for improvement. But the overall article is too lengthy, and I suggest making appropriate revisions to focus solely on topics related to the topic. For example, P2's "2. Sensor properties of faba bean constituents" and P4's "3 Faba bean flavor and its beany flavor Formation”, I suggest deleting these two parts or simply summarizing the results of existing literature research reports, as they will be discussed later when introducing the impact of processing processes on the quality of broad beans (including physics, texture, flavor, etc.).
Author Response
Recommended sections for removal has been deleted.